# Optimization of a Photobiomodulation Protocol to Improve the Cell Viability, Proliferation and Protein Expression in Osteoblasts and Periodontal Ligament Fibroblasts for Accelerated Orthodontic Treatment

**DOI:** 10.3390/biomedicines12010180

**Published:** 2024-01-14

**Authors:** Aline Gonçalves, Francisca Monteiro, Sofia Oliveira, Inês Costa, Susana O. Catarino, Óscar Carvalho, Jorge Padrão, Andrea Zille, Teresa Pinho, Filipe S. Silva

**Affiliations:** 1UNIPRO—Oral Pathology and Rehabilitation Research Unit, University Institute of Health Sciences (IUCS), CESPU, 4585-116 Gandra, Portugal; aline.goncalves@iucs.cespu.pt (A.G.); iamalvescosta@gmail.com (I.C.); teresa.pinho@iucs.cespu.pt (T.P.); 2Center for MicroElectroMechanical Systems (CMEMS), University of Minho, Campus Azurém, 4800-058 Guimarães, Portugal; sofiaoliveira@dem.uminho.pt (S.O.); scatarino@dei.uminho.pt (S.O.C.); oscar.carvalho@dem.uminho.pt (Ó.C.); fsamuel@dem.uminh.pt (F.S.S.); 3ICVS/3B’s-PT Government Associate Laboratory, 4710-057 Braga, Portugal; 4LABBELS—Associate Laboratory, 4710-057 Braga, Portugal; 5Centre for Textile Science and Technology (2C2T), Department of Textile Engineering, University of Minho, Azurém Campus, 4800-058 Guimarães, Portugal; padraoj@2c2t.uminho.pt (J.P.); azille@det.uminho.pt (A.Z.); 6IBMC—Instituto Biologia Molecular e Celular, i3S—Instituto de Inovação e Investigação em Saúde, Universidade do Porto, 4200-135 Porto, Portugal

**Keywords:** bone remodeling, fibroblasts, optimization, osteoblasts, photobiomodulation

## Abstract

Numerous pieces of evidence have supported the therapeutic potential of photobiomodulation (PBM) to modulate bone remodeling on mechanically stimulated teeth, proving PBM’s ability to be used as a coadjuvant treatment to accelerate orthodontic tooth movement (OTM). However, there are still uncertainty and discourse around the optimal PBM protocols, which hampers its optimal and consolidated clinical applicability. Given the differential expression and metabolic patterns exhibited in the tension and compression sides of orthodontically stressed teeth, it is plausible that different types of irradiation may be applied to each side of the teeth. In this sense, this study aimed to design and implement an optimization protocol to find the most appropriate PBM parameters to stimulate specific bone turnover processes. To this end, three levels of wavelength (655, 810 and 940 nm), two power densities (5 and 10 mW/cm^2^) and two regimens of single and multiple sessions within three consecutive days were tested. The biological response of osteoblasts and periodontal ligament (PDL) fibroblasts was addressed by monitoring the PBM’s impact on the cellular metabolic activity, as well as on key bone remodeling mediators, including alkaline phosphatase (ALP), osteoprotegerin (OPG) and receptor activator of nuclear factor κ-B ligand (RANK-L), each day. The results suggest that daily irradiation of 655 nm delivered at 10 mW/cm^2^, as well as 810 and 940 nm light at 5 mW/cm^2^, lead to an increase in ALP and OPG, potentiating bone formation. In addition, irradiation of 810 nm at 5 mW/cm^2^ delivered for two consecutive days and suspended by the third day promotes a downregulation of OPG expression and a slight non-significant increase in RANK-L expression, being suitable to stimulate bone resorption. Future studies in animal models may clarify the impact of PBM on bone formation and resorption mediators for longer periods and address the possibility of testing different stimulation periodicities. The present in vitro study offers valuable insights into the effectiveness of specific PBM protocols to promote osteogenic and osteoclastogenesis responses and therefore its potential to stimulate bone formation on the tension side and bone resorption on the compression side of orthodontically stressed teeth.

## 1. Background

The therapeutic potential of photobiomodulation (PBM) has been vastly described, being currently employed for multiple biomedical applications. By now, a plethora of evidence supports that certain PBM protocols may modulate bone remodeling phenomena on mechanically stimulated teeth, demonstrating PBM’s ability to aid the acceleration of orthodontic tooth movement (OTM) [1,2,3]. However, there are still uncertainty and discourse about the most appropriate PBM protocols [4,5,6,7], which hampers its optimal and well-established clinical applicability. This is a consequence of the poor translatability from cellular to animal models and then to clinical practice. Therefore, combining the efforts of basic research and systematic optimization processes is pivotal to achieve a solid translation of the PBM of OTM to a clinical context.

The mechanism of action of PBM involves the absorption of photons by the cytochrome c oxidase (CCO), leading to the stimulation of adenosine triphosphate (ATP) production, augmented nitric oxide (NO) release and increased DNA and RNA synthesis and repair [2,8,9]. Interestingly, PBM was found to produce an acute and transient generation of intracellular reactive oxygen species (ROS) in normal, non-stressed cells. However, a reduction in ROS levels following irradiation was observed in oxidatively stressed cells or in animal models of disease [8,9], revealing the potential of PBM to differentially stimulate stressed and/or diseased cells and tissues so as to restore cellular homeostasis and normal activity. Altogether, these effects stimulate the proliferation, differentiation and activity of the mechanically stressed cells and tissues of the periodontium, promoting several cellular biological phenomena implicated in OTM, including improved regeneration and remodeling, root resorption control and angiogenesis [4,10,11,12]. Moreover, the expression of specific bone remodeling mediators can be also modulated using PBM; different studies show the potential of PBM to impact the activity of alkaline phosphatase (ALP), osteoprotegerin (OPG) and the receptor activator of nuclear factor κ-Β ligand (RANK-L), amongst other things, in favor of osteoclastogenesis or osteogenesis, depending on the PBM parameters [10,13,14]. Given the differential metabolic patterns exhibited in the tension and compression sides of orthodontically stressed teeth [15,16], it seems recognizable that PBM should be customized and adapted to modulate fibroblastic and osteoblastic activity and stimulate bone remodeling. Specifically, on the tension side, bone formation needs to be fostered to counteract the space left by tooth movement, while osteoclastogenesis (i.e., bone resorption) must predominate on the compression side, where the moved teeth will occupy a space previously covered by bone [15,17]. Hence, different types of irradiation may be applied to each side of the teeth. In this sense, we propose an optimization protocol to find the most appropriate PBM parameters to stimulate specific bone turnover processes.

Periodontal ligament (PDL) fibroblasts and osteoblasts are the most important cells involved in bone remodeling processes, the modulation of inflammatory responses and protection against tissue damage and homeostatic imbalance in the periodontal space [15,18]. In the end, these are the cells that regulate the entire remodeling and regenerative processes required for tooth movement to happen, and thereby must be the main PBM targets. Indeed, previous in vitro studies showed that PBM is able to modulate the proliferation and gene expression of osteoblasts [19,20,21], as well as induce osteoblastic or osteoclastic differentiation in fibroblasts [16] and PDL stem cells [20,22,23]. However, the variability among the PBM parameters is huge, and the optimal stimulation protocols are still undefined. In this sense, the impact of different PBM protocols on the cellular metabolic activity of PDL fibroblasts and osteoblasts, as well as on the expression of key bone remodeling mediators, including ALP, OPG and RANK-L, was addressed daily over three consecutive days. Based on this evidence, this study will allow the settlement of the optimal PBM protocols to stimulate specific bone remodeling processes, aiding the future consolidation of PBM as a coadjuvant treatment to accelerate OTM.

## 2. Material and Methods

### 2.1. Cell Culture

The human fetal osteoblast cell line (hFOBs) was purchased from the American Type Culture Collection (CRL-11372, ATCC^®^, Manassas, VA, USA). The cells were maintained in Dulbecco’s Modified Eagle Medium/Nutrient Mixture F-12 (DMEM:F12, 1:1) without phenol red (PAN-Biotech GmbH, Aidenbach, Germany), containing 10% (*v*/*v*) fetal bovine serum (FBS, Sigma-Aldrich, St. Louis, MO, USA), 2.5 mM L-glutamine (PAN-Biotech^®^, Germany) and 0.3 mg/mL antibiotic G418 (PAN-Biotech GmbH, Germany).

The human periodontal ligament fibroblasts (hPLFs) were acquired from Innoprot^®^ (P10867, Innoprot^®^, Derio, Spain). The cells were cultured in culture medium composed of DMEM:F12 Mix (1:1) with stable glutamine and 1.2 g/L NaHCO_3_ (PAN-Biotech^®^, Germany), supplemented with 1% (*v*/*v*) penicillin/streptomycin and 10% (*v*/*v*) FBS.

Both the osteoblasts and fibroblasts were maintained in an incubator at 37 °C and 5% (*v*/*v*) CO_2_, in a humidified atmosphere of 100% relative humidity. The medium of both cell lines was changed twice a week, and both cells were observed every day using an inverted microscope (Kern^®^, Lohmar, Germany) at 10× magnification to detect morphologic changes.

### 2.2. Cell Seeding

When the confluence reached 80–90%, both cell lines were trypsinized with 0.25% (*v*/*v*) trypsin/EDTA and centrifuged at 300× *g* for 5 min. The hFOBs and hPLFs were seeded in 12-well plates at a concentration of 20,000 cells/mL. The hFOBs and hPLFs were used at passage numbers 12–14 and 2–3, respectively.

Each 12-well plate was tested with different PBM conditions to avoid any interference between the experimental conditions. Both cells were stimulated 72 h after seeding.

### 2.3. Photobiomodulation (PBM) Protocol

A stimulation device comprising light-emitting diodes (LEDs) of 655, 810 and 940 nm was designed and built by the authors’ research group—the Center for Microelectromechanical Systems (CMEMS)—(see Figure 1). The stimulation device was connected to a computer, which controlled the LEDs operation, namely their activation/deactivation, delivered power density and stimulation time.

The PBM stimulation was conducted in two phases. First, an initial screening of a variety of PBM conditions utilized in the existing literature was conducted to assess the effect on the cells’ metabolic activity. For each LED, two power densities were tested: 5 mW/cm^2^ and 10 mW/cm^2^. The PBM stimulation was performed in continuous mode (0 Hz) once for 3 min and cellular analyses were performed after 1, 24 and 72 h. The most promising PBM conditions were selected and applied in a second stimulation stage—the optimal parameter testing phase—consisting of daily stimulations provided over three consecutive days to investigate the effect of multiple stimulation sessions (see Figure 2). After each 3 min stimulation, cellular analyses were performed, and the culture medium was collected to undergo protein expression assays. A summary of the PBM parameters addressed here is provided in Table 1.

A control (non-irradiated) group was established by placing the turned-off stimulation device on top of the respective wells for 3 min. This assured that all experimental groups were subjected to the same conditions, allowing reliable comparison. Figure 2 depicts the experimental design established for the study.

### 2.4. Cellular Assays

#### 2.4.1. Metabolic Activity

To assess cellular metabolic activity, 3-(4,5-dimethylthiazol-2-yl)-5-(3-carboxymethoxyphenyl)-2-(4-sulfophenyl)-2H-tetrazolium (MTS) dye is a colorimetric method used. MTS is a tetrazolium compound that is reduced by metabolically active cells, leading to the production of a formazan product, soluble in culture medium, that is quantified colorimetrically by measuring its absorbance at 490 nm. This is directly proportional to the amount of metabolically active cells in the culture [24] and assures a reliable and quantitative method to compare the cell viability among groups.

The metabolic activity of the hFOBs and hPLFs was assessed using an MTS assay (Abcam^®^, Cambridge, UK), according to the manufacture’s protocol, 1, 24 and 72 h after stimulation during the initial screening. Then, in the optimal parameter testing phase, the metabolic activity was addressed one, two and three days after the onset of the PBM stimulation (everyday stimulation for three days). The blank was performed by adding the MTS reagent to wells containing only culture medium (without cells). The cells and the blank (n = 4) were incubated for 4 h at 37 °C, 5% (*v*/*v*) CO_2_ and 100% relative humidity. After incubation, the volume of each well was read in triplicate in a 96-well plate using a microplate reader (BioTek^®^ Epoch, Burlingame, CA, USA) at 490 nm.

#### 2.4.2. Cell Counting

In the second stage of stimulation, the optimal parameter testing phase, to quantify the cell proliferation, 0.4% (*v*/*v*) trypan blue (PAN-Biotech GmbH, Germany) was used, and the cells were counted using a Neubauer chamber. Trypan blue allows us to determine the number of viable cells as it dyes dead cells (membrane integrity compromised) with a distinct blue color when observed under a microscope, while living cells appear uncolored (intact cell membrane). The results of this cell proliferation assay were used to normalize the remaining cellular assays, which are described below.

#### 2.4.3. Alkaline Phosphatase Expression

During the optimal parameter testing phase, after one, two and three days of PBM, the culture medium was removed, and the ALP activity was quantified using the colorimetric Alkaline Phosphatase Assay Kit (Biovision^®^, Piscataway, NJ, USA), following the manufacturer’s instructions. This kit uses p-nitrophenyl phosphate (pNPP) as a phosphatase substrate that turns yellow when dephosphorylated in the presence of ALP. Briefly, 80 μL of culture medium of each condition, control and blank was incubated with 5 mM of pNPP solution for 1 h at 25 °C, protected from light. The standard curve was prepared according to the manufacturer´s protocol. Then, a stop solution was added into each standard and sample, and the optical density was read at 405 nm using a microplate reader (BioTek^®^ Epoch, USA).

#### 2.4.4. Enzyme-Linked Immunosorbent Assay (ELISA) Assays

The expression of soluble OPG and RANK-L was quantified in the second phase of stimulation by performing an enzyme-linked immunosorbent assay (ELISA) on the cell culture medium for each condition. For this purpose, a human OPG ELISA kit (Elabscience^®^, Houston, TX, USA) and a human RANK-L ELISA kit (Elabscience^®^, USA) were used. The assays were performed according to the manufacturer’s indications, as well as the preparation of the reagents. Briefly, the culture medium samples and standards were added in duplicate to a 96-well ELISA plate coated with an antibody specific to human-soluble OPG or RANK-L for 90 min at 37 °C. Then, the detection antibody and horseradish peroxidase (HRP) conjugate were successively incubated for 1 h and 30 min, respectively, at 37 °C, followed by the substrate reagent for 15 min at 37 °C, protected from light. The stop solution was added to each well and the absorbance was immediately read at 450 nm using a microplate reader (BioTek^®^ Epoch, USA). The concentration of human OPG and RANK-L in the samples was calculated according to the manufacturer’s indications.

### 2.5. Statistical Analysis

The statistical analysis and graphical production were performed in GraphPad^®^ version 8.0.2. The Shapiro–Wilk test was used to assess the normality of the data. For data that were normally distributed, one-way ANOVA was used, followed by Tukey’s HSD test for multiple comparisons. When data were not normally distributed, the non-parametric Kruskal–Wallis test was used, followed by Tukey’s HSD test for multiple comparisons. The Cohen’s q value was used to evaluate the effect size among groups. For all statistical analyses, the significance level was set at 0.05, and the results were expressed as mean ± standard deviation.

## 3. Results

### 3.1. Initial Screening

The colorimetric MTS assay was used to assess the cellular metabolic activity. At the first timepoint (1 h after irradiation), PBM did not affect the metabolic activity of the hPLFs (Figure 3A). In the groups irradiated with 810 nm (5 and 10 mW/cm^2^) and 940 nm at 5 mw/cm^2^, 24 hours after stimulation, the hPLFs elicited increased metabolic activity compared to the control group (*p*-values = 0.0178, 0.0262, 0.0170, respectively). However, this augmented metabolic activity in relation to the control group was reversed at 72 h after irradiation, in which no statistically significant differences were observed between the metabolic activity of the irradiated cells and the control.

For the hFOBs, the groups irradiated with 655 nm at 10 mW/cm^2^ and with 810 and 940 nm at both levels of power density (5 and 10 mW/cm^2^) showed greater metabolic activity 1 h after stimulation compared to the control group (*p*-values = 0.0339; <0.0001; 0.0300; <0.0001; 0.0011, Figure 3B). At the 24 h follow-up, irradiation had not induced significant alterations in the metabolic activity of the hFOBs. At the third follow-up (72 h after irradiation), the groups receiving irradiation of 655 nm (5 and 10 mW/cm^2^) and 940 nm (5 and 10 mW/cm^2^) showed a significant reduction in metabolic activity compared to the controls (*p*-values = 0.0003, 0.0174, 0.0002, 0.0003, respectively).

### 3.2. Optimal Parameter Testing Phase

After the initial screening phase, which envisioned the selection of the most effective PBM parameters to promote osteoblast and fibroblast metabolic activity, the cells were daily stimulated over three consecutive days to assess the effect of prolonged exposure to light stimuli on the cells’ activity, as well as on protein expression (ALP, OPG and RANK-L). The best conditions for each wavelength were selected according to the statistical significance and effect size. When no statistically significant differences existed, the conditions depicting a greater effect size on the cell viability at the last follow-up (72 h after stimulation) were selected.

For the optimal parameter testing phase, data were normalized by cell concentration, to ensure that a differential cell proliferation among groups would not mask the results. Thus, the cell concentration addressed using trypan blue was assessed at all timepoints (days 1, 2 and 3 after light stimulation).

#### 3.2.1. Metabolic Activity

Concerning the effect of PBM on the metabolic activity of the hPDLs, there was no statistically significant differences between any of the groups at all timepoints (Figure 4A). The metabolic activity of cells irradiated with 655 nm at 10 mW/cm^2^ showed a slight increase in relation to the control by day 2, although it was not statistically significant. The cellular activity tended to decrease from the first to the last day of stimulation in all groups.

The metabolic activity of the hFOBs followed a similar trend, eliciting maximum activity after the first irradiation, while the minimum metabolism rate was found at day 3 (Figure 4B). On day 1, the osteoblasts stimulated with light of 810 nm delivered at 5 mW/cm^2^ exhibited a significantly higher metabolic activity than the controls (*p*-value < 0.0001). This augmentation withered after the second stimulation, but it was restored on day 3 (*p*-value = 0.0144). Similarly, a significant increase in the metabolic activity of the osteoblasts was observed after the last stimulation on the 655 nm group (*p*-value = 0.0124). These results suggest that PBM with 655 nm applied at 10 mW/cm^2^ and 810 nm at 5 mW/cm^2^ delivered daily over three days is capable of stimulating the metabolic activity of osteoblasts but does not modulate the activity of PDL fibroblastic cells.

#### 3.2.2. Alkaline Phosphatase Expression

ALP is a biomarker of early osteoblastic differentiation commonly used as a proxy for bone formation [23,25]. Here, the ALP expression in the hPLF cells decreased over time for the control and experimental groups, except for the cells irradiated with 810 nm at 5 mW/cm^2^, which presented a slight, non-significant increase in ALP release on day 2, although it had returned to levels comparable to the remaining groups after the third day of stimulation. The hPDLs irradiated with 655 nm at 10 mW/cm^2^ elicited increased metabolic activity on day 2 compared to the control group (*p*-value = 0.0009), while no statistically significant differences were observed for the remaining PBM protocols (Figure 5A).

In the case of the hFOBs, a similar downward trend in ALP expression was observed over the three days of stimulation for all groups. In parallel with the augmented metabolic activity profile induced in the osteoblasts (Figure 5B), a single PBM session of 810 nm at 5 mW/cm^2^ induced a significant increase in ALP expression in relation to the control (*p*-value = 0.0088). In addition, the metabolic activity of the osteoblastic cells irradiated with 655 nm at 10 mW/cm^2^, 810 nm at 5 mW/cm^2^ and 940 nm at 5 mW/cm^2^ was also increased by the third day of stimulation (*p*-value = <0.0001, <0.0001, 0.0014, respectively). This evidence demonstrates the potential of PBM to promote new bone formation by stimulating the ALP activity in both osteoblastic and fibroblastic cells.

#### 3.2.3. Osteoprotegerin (OPG) and Receptor Activator of Nuclear Factor κ-Β Ligand (RANK-L) Expression

OPG and RANK-L are key players in the bone remodeling process that occurs during tooth movement. While RANK-L is a bone resorption mediator that is typically upregulated on the compression side of the teeth, OPG is a biomarker commonly used to monitor bone formation on the tension side of mechanically stressed teeth [25,26]. OPG prevents excessive bone resorption by binding to RANK-L, hindering its association with RANK and therefore blocking the bone resorption cascade [26]. These alterations in the cell remodeling pattern are underlain by the activity of the osteoclasts (responsible for destroying bone), which are mainly regulated by the balance between RANK-L, OPG and RANK.

Here, the OPG concentration for the hPLFs decreased over time in all groups, including the controls (Figure 6A). The group irradiated daily with 655 nm at 10 mW/cm^2^ showed a significantly higher OPG expression at all timepoints (*p*-values = 0.0011, 0.0002, 0.0137, respectively), suggesting that these PBM parameters are particularly suitable to promote osteoblastic differentiation of the hPLFs.

On the other hand, the hFOBs revealed a distinct pattern of OPG expression among groups (Figure 6B); light of 810 nm was able to significantly increase the OPG expression after a single stimulation (*p*-value = 0.0313), although this effect had been reversed by day 2, when a significantly diminished OPG concentration was observed (*p*-value = 0.0246). However, on day 3, all the irradiated groups exhibited increased OPG expression compared to the control group, including the hFOBs irradiated with 810 nm at 5 mW/cm^2^ (*p*-values = <0.0001, <0.0001, 0.0035, respectively). Together with the ALP expression data, these results support the effectiveness of PBM in potentiating the expression of bone formation mediators.

The expression of RANK-L was also assessed using ELISA. However, the variability within groups was huge, disabling reliable comparisons. No statistically significant differences were found between the irradiated and control groups at any timepoint, neither in the hPLFs nor the hFOBs. Overall, the hPLFs showed a lower RANK-L expression than the hFOBs, as expected. Figure 7A suggests that 810 nm PBM may inhibit the RANK-L expression of hPLFs after the first stimulation, but all the irradiation protocols lead to augmented, non-significant RANK-L expression by day 2 compared to the control. Also, non-irradiated hFOBs and hFOBs irradiated with 940 nm delivered at 5 mW/cm^2^ showed a residual RANK-L expression over the three stimulation days. Highly heterogeneous data were obtained for the other two groups, and so no further assumptions about then RANK-L expression of hFOBs under irradiation can be made (Figure 7B).

Then, the RANK-L/OPG ratio was calculated. Given the great variability in the RANK-L data, and since the respective errors would be propagated and amplified over the calculations, the RANK-L/OPG ratio data were not considered for analysis.

The main outcomes of the current optimization protocol are depicted in Table 2.

## 4. Discussion

PBM promotes cellular activity and consequently effects cell viability, proliferation and growth via the stimulation of the mitochondrial respiratory chain, resulting in enhanced mitochondrial membrane potential and increased ATP synthesis [3,4,8,11,20,27]. This ability to modulate cellular dynamics has been used in different areas of biomedicine for several years [1,28,29]. In orthodontics, its effectiveness in accelerating OTM is based on its ability to regulate bone remodeling mediators [1,30,31,32,33]. So far, the most commonly used wavelengths range from 600 to 1200 nm, with special emphasis on the potential of the segments of 650 ± 30 nm, 810 ± 40 nm and 940 ± 40 nm to promote bone remodeling phenomena [3,16,18,34,35,36]. Irradiation delivered at low power densities (i.e., the measure of the light power delivered to cells per area) in the range of 1.5 to 15 mW/cm^2^ has been proved to modulate various cytological pathways in different cell types, including periodontal cells [18,37,38], osteoblasts [18,20,36], chondrocytes [29] and neuronal cells [28], among others. PBM parameters such as the type of laser, intermittence (i.e., continuous or pulsing mode), duration of irradiation and periodicity vary significantly among experiments. In this sense, the current experimental work attempts to fill the gap in the fundamental basis of PBM’s application in orthodontics: the settlement of the most effective parameters to stimulate the target cells in the periodontal tissue. To this end, an LED stimulation device was programmed to irradiate at three different wavelengths (i.e., 655, 810 and 940 nm) and two power densities (i.e., 5 and 10 mW/cm^2^), under a regimen of single and multiple sessions within three consecutive days, monitored on each day.

During the first phase of the present study, the initial screening, cells were exposed to various PBM regimens, as shown in Table 1. Figure 3A depicts an increase in the metabolic activity of the hPLFs irradiated with 810 nm at 5 mW/cm^2^, 810 nm at 10 mW/cm^2^ and 940 nm at 5 mW/cm^2^ at the 24 h follow-up compared to the control, which withered after 72 h (Figure 3A). PBM produced a similar effect on the osteoblasts’ metabolic activity. Although a significant increase in the hFOBs’ viability was observed 1 h after PBM of 655 nm at 10 mW/cm^2^, 810 nm (5 and 10 mW/cm^2^) and 940 nm (5 and 10 mW/cm^2^) compared to the control, this acute effect was dissolved after 24 h and reversed after 72 h, as some irradiated groups revealed a significant decrease in metabolic activity at this last timepoint (655 and 940 at both 5 and 10 mW/cm^2^) (Figure 3B). Similar observations were reported by Bölükbaşı Ateş, Can and Gülsoy (2017); the authors observed a statistically significant increase in the osteoblasts’ viability (evaluated using MTT assay, a cell activity bioassay similar to MTS) after irradiation, but the effect did not last up to 72 h [19]. Based on these data, different levels of irradiation have a positive acute and transient effect on the metabolic activity and viability of both hPLFs and hFOBs, returning to the control levels after a few hours. This supports the necessity for periodic PBM sessions to produce a long-lasting effect on fibroblastic and osteoblastic cells.

From the initial screening, the most suitable combinations of PBM parameters to promote cellular activity were selected. The parameters that most favorably affected the cell metabolism were the same for the hPLFs and hFOBs, namely 655 nm at 10 mW/cm^2^, 810 nm at 5 mW/cm^2^ and 940 nm at 5 mW/cm^2^. At this point, the optimal parameter testing phase began; three stimulation sessions were delivered over three consecutive days, and the metabolic activity and protein expression data were normalized by cell concentration to consider the effect of consecutive PBM sessions on the cell count over time. First, the metabolic activity of the hPLF and hFOB cells was addressed again; all the groups (irradiated and non-irradiated) elicited a similar downward trend in their cell activity over time (Figure 4), which means that the increase in the number of cells was not accompanied by an increase in the overall metabolic activity over the three stimulation days. No statistically significant differences in the normalized metabolic activity of the hPLFs were found after day 1, 2 and 3 (Figure 4A). On the contrary, a significant increase in the hFOBs’ metabolic activity was observed after PBM of 810 nm applied at 5 mW/cm^2^ on day 1, and after three irradiations of 655 nm at 10 mW/cm^2^ and 810 nm at 5 mW/cm^2^ on day 3 (Figure 4B). These results are in line with the reports by Bölükbaşı Ateş and Can (2017) and Chaweewannakorn et al. (2021), who observed a PBM-driven augmentation in the cellular activity of the osteoblasts (daily irradiation, 830 nm, 3 mW/cm^2^, after days 6 and 8) and PDL stem cells (single session, 810 nm, 50 mW/cm^2^, after 48 and 72 h), respectively [19,20]. As far as we know, this is the first study investigating the effect of PBM on the metabolic activity of PDL fibroblasts. Based on the evidence provided by the present study and former reports [19,20], daily PBM sessions are able to consistently enhance the osteoblastic activity over three consecutive days, which may be used to stimulate osteogenic pathways in orthodontically stressed teeth.

Then, the impact of PBM on the cell differentiation was addressed by assessing the expression of specific bone remodeling mediators, namely ALP, OPG and RANK-L. In the present study, all the groups of hPLFs and hFOBs depicted decreasing ALP expression over time, including the control groups (Figure 5), which suggests that these cells may naturally lose their osteoblastic phenotype over the days that follow their seeding [15]. Importantly, PBM of 655 nm at 10 mW/cm^2^ was able to upregulate the ALP expression of the hPLFs on day 2 in relation to the control, although it had returned to control levels after the third irradiation (Figure 5A). Nevertheless, these data show the potential of light irradiation with the indicated properties to further the osteoblast-like characteristics of hPLFs [15,39].

In addition, the hFOBs exposed to all the PBM protocols elicited an ALP overexpression after the third stimulation (day 3) compared to the control group (Figure 5B), and 810 nm light was able to increase the hFOBs’ ALP expression right after the first stimulation. This evidence suggests that light can actually stimulate osteoblastic differentiation and activity and consecutively promote bone formation in the periodontal space, corroborating previous reports [21,40]. Importantly, even without the application of a mechanical load, which triggers the osteoblastic-like characteristics in fibroblasts [15,39], PBM was able to stimulate the expression of ALP in the hPLFs, potentiating their ability to form new bone matrices [14,15,39,41]. Overall, several pieces of evidence point to the ability of different levels of PBM to improve the ALP expression and activity in both osteoblastic and fibroblastic cells, which should be stimulated to induce the maximum bone formation. Enhancing this phenomenon is particularly important on the tension side of the teeth subjected to orthodontic forces applied in the mesiodistal and/or buccolingual directions when translation movements are required.

Finally, to complement the analysis of bone remodeling patterns following exposure to PBM, the impact of PBM on the expression of OPG and RANK-L was assessed using ELISA. These are key players in the bone remodeling process that occurs during tooth movement. The ability of PBM to upregulate the expression of OPG in osteoblastic cultures has already been described [21,42], but not in fibroblasts. In the current study, all the hPLF groups showed a downward trend in OPG expression over time, as observed for the ALP protein, suggesting a natural downturn in the osteoblastic phenotype. However, the hPLFs treated with 655 nm at 10 mW/cm^2^ revealed a significantly higher OPG concentration than the controls at all timepoints (Figure 6A), suggesting a great potential of this daily PBM protocol to promote the osteoblastic differentiation of the hPLFs.

Similar to what was observed for the hPLFs, the hFOBs exhibited a consistent decrease in OPG expression over time (Figure 6B). Importantly, all the tested PBM protocols elicited an upregulation of OPG expression by day 3, demonstrating that all three combinations of the PBM parameters tested here may promote bone formation by increasing OPG expression. Future studies must increase the evaluation span to ascertain whether the longer light stimulation protocols would continue to have a positive, long-lasting impact on OPG and ALP expression and other bone remodeling mediators. If the PBM-induced increase in OPG and ALP expression remains for more than three days, that would mean that PBM is able to change the RANK-L/OPG ratio in favor of osteogenesis on the tension side, and thereby counteract the effect of orthodontic mechanical stress, which promotes hypoxia and ischemia of the tissues, favoring osteoclastogenesis. Although the expression of RANK-L was assessed using ELISA in the present study (Figure 7), the heterogeneity of the data within each experimental group was too great to produce reliable inferences, also disabling the calculation of the RANK-L/OPG ratio. Nevertheless, the hPLFs elicited a slight, non-significant increase in RANK-L expression by day 2 after 810 nm irradiation compared to the control. Together with the OPG data, this may be an indication that two PBM sessions delivered on consecutive days could induce the osteoclastogenic differentiation of hPLFs by stimulating RANK-L expression, while a third stimulation may benefit the osteogenic pathways by augmenting the OPG expression in hFOBs. The literature presents some conflicting reports on this topic; some studies observed that, along with the upregulation of OPG expression confirmed here, PBM is able to reduce the RANK-L expression and protein levels in osteoblasts [21,42] and osteoclast/osteoblast co-cultures [21], resulting in a significant reduction in the RANK-L/OPG ratios induced by irradiation, thus promoting bone formation [21,42]. On the other hand, others have described an increase in RANK-L expression in osteoblasts subjected to 1064 nm pulsed irradiation (20–30 Hz), which favored the RANK-L/OPG ratio, and hence osteoclastogenesis [43]. Importantly, when subjected to orthodontic-mimicking forces, Tabatabaei and colleagues (2023) found that the activity of PDL fibroblasts was modulated by 980 nm PBM toward an upregulation of bone resorption genes (including RANK-L) and a downregulation of bone formation mediators under compressive load, while the opposite was observed when the cells were under tensile forces [16]. This evidence suggests that the inconsistent reports about the effect of PBM on OPG and RANK-L expression may be a consequence of how this response depends on the external conditions of the periodontal tissues, such as mechanical stress.

Altogether, the evidence presented in the current study demonstrates the effectiveness of daily PBM of 655 nm delivered at 10 mW/cm^2^ and 810 nm at 5 mW/cm^2^ delivered for three consecutive days to modulate the periodontal environment to favor osteogenesis via distinct osteogenic pathways that involve the upregulation of bone formation mediators, such as ALP and OPG, and the stimulation of osteoblastic differentiation, proliferation and growth. Furthermore, this study showed that PBM of 810 nm at 5 mW/cm^2^ delivered for two consecutive days, but not applied on the third day, is able to downregulate OPG expression and produce a slight non-significant increase in RANK-L expression, favoring osteoclastogenesis and thus bone resorption.

It is worth noting that the current study presents some limitations; first of all, only two independent samples were used to perform the ELISA assays for the determination of the OPG and RANK-L concentration after irradiation. Although the OPG expression has shown low fluctuations for each test condition, the variability in the RANK-L data was considerably higher. This hindered the evaluation of the impact of PBM on the expression of this key bone resorption mediator, which subsequently disabled the calculation of the RANK-L/OPG ratio. Future studies must consider a minimum of three independent samples for quantification of the RANK-L concentration using ELISA.

## 5. Conclusions

Since the discovery of the potential of PBM to modulate bone remodeling and to accelerate dental movement, several efforts have been dedicated to the clinical implementation of light stimulation protocols as a coadjuvant to orthodontic treatment. However, the lack of well-established PBM protocols has hampered its consolidation has a reliable acceleration technique among orthodontists, which is a consequence of the poor translatability from cellular to animal models. The current in vitro study has defined the optimal PBM parameters to enhance fibroblastic and osteoblastic metabolism and differentiation:On the tension side, daily PBM of 655 nm delivered at 10 mW/cm^2^, 810 nm at 5 mW/cm^2^ and 940 nm at 5 mW/cm^2^ should be applied for three consecutive days to stimulate bone formation;On the compression side, PBM of 810 nm at 5 mW/cm^2^ must be applied for two consecutive days, and the stimulation must be suspended by the third day in order to promote bone resorption phenomena;Low irradiation intensity levels (up to 10 mW/cm^2^) are enough to produce positive bioeffects on fibroblastic and osteoblastic metabolism.

## 6. Future Perspectives

Even though this experimental work provides fundamental knowledge on the PBM protocols that may be used to modulate bone remodeling in specific periodontal areas, there is no optimal PBM parametrization to assure a robust and well-established acceleration of orthodontic tooth movement in a clinical context yet. Instead, this work will be a launching ramp for forthcoming developments on the PBM protocols to be employed in animal studies, which could be then reliably translated to clinical practice. This study highlights the importance of the following points to be considered in future animal studies:Longitudinal analysis may be held to ascertain the long-term responsiveness to irradiation, including protein expression patterns and safety concerns (e.g., tissue damage);Different stimulation periodicities must be tested (e.g., cycles of two stimulation days/one suspension day, sessions every other day, sessions every two days), as daily stimulation may not be ideal for enhancing the metabolism of certain periodontal tissues;The monitorization of inflammatory markers, such as IL-1β and other cytokines, under PBM may be considered, since pro-inflammatory markers modulate the activity and differentiation of periodontal ligament fibroblastic cells, which are pivotal in bone turnover mechanisms and soft tissue regeneration;Given the influence of external conditions, such as the biochemical environment and mechanical loads, on the effect of PBM on the expression of key bone remodeling mediators, future in vitro and animal studies should focus on investigating how the optimal PBM protocols defined here can impact the biodynamics of cells and tissues subjected to tensile and compressive forces;Different physical stimuli, such as light, mechanical deformations, electrical signals and others, are capable of activating and modulating distinct metabolic and signaling pathways in the periodontal tissues. In this sense, the combination of PBM with alternative therapeutic modalities (e.g., ultrasound stimulation, application of static magnetic fields) may be an interesting feature to unveil in forthcoming research.

## Figures and Tables

**Figure 1 biomedicines-12-00180-f001:**
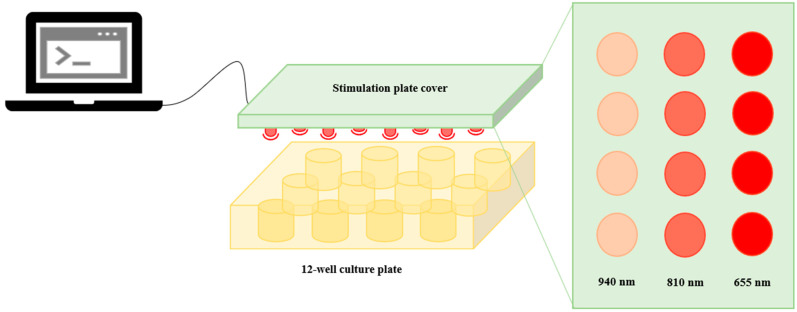
Schematic representation of the light stimulation setup.

**Figure 2 biomedicines-12-00180-f002:**
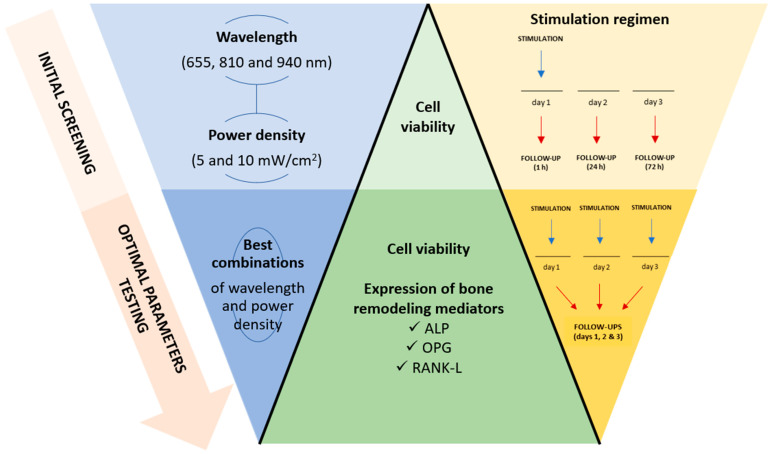
Experimental design and plan of the study.

**Figure 3 biomedicines-12-00180-f003:**
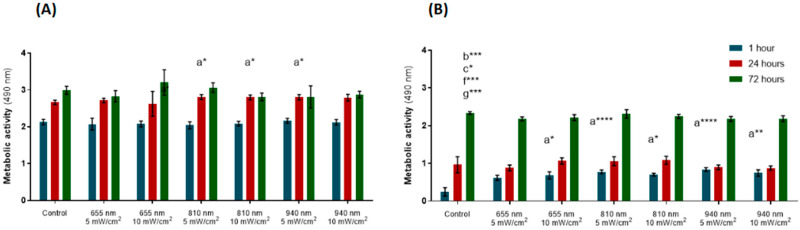
Initial screening of photobiomodulation parameters through the evaluation of metabolic activity using the MTS assay. (**A**) Human periodontal fibroblasts (hPLFs). (**B**) Human femoral osteoblasts (hFOBs). Statistical differences were denoted as a for control; b for 655 nm, 5 mW/cm^2^; c for 655 nm, 10 mW/cm^2^; f for 940 nm, 5 mW/cm^2^; and g for 940 nm, 10 mW/cm^2^. Statistically significant differences in relation to control are indicated above the highest metabolic activity value. * *p*  <  0.05; ** *p*  <  0.01; *** *p*  <  0.001; **** *p*  <  0.0001. Data are represented as mean ± SD (n = 4).

**Figure 4 biomedicines-12-00180-f004:**
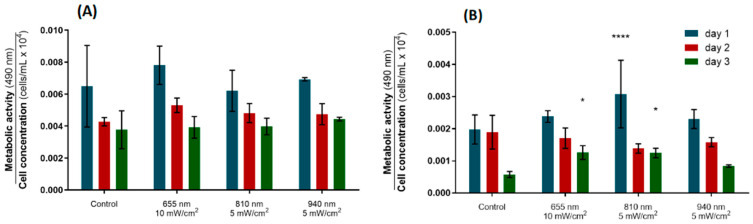
Metabolic activity data (MTS assay) for the optimal photobiomodulation parameters, normalized by cell counting. (**A**) Normalized OPG concentration for hPLFs. (**B**) Normalized OPG concentration for hFOBs. Statistically significant differences in relation to control are indicated above the highest metabolic activity value. * *p*  <  0.05; **** *p*  <  0.0001. Data are represented as mean ± SD (n = 4).

**Figure 5 biomedicines-12-00180-f005:**
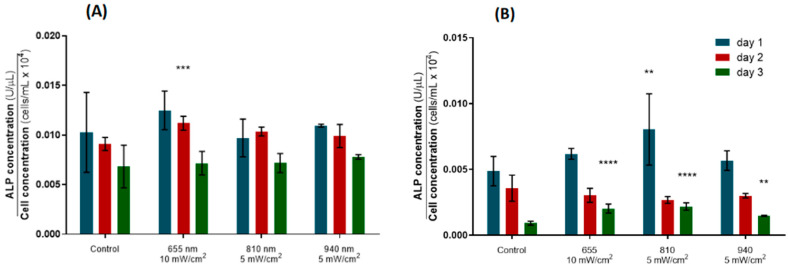
ALP expression data (colorimetric assay) for the optimal photobiomodulation parameters, normalized by cell counting. (**A**) Normalized ALP concentration for hPLFs. (**B**) Normalized ALP concentration for hFOBs. Statistically significant differences in relation to control are indicated above the highest ALP concentration value. ** *p*  <  0.01; *** *p*  <  0.001; **** *p*  <  0.0001. Data are represented as mean ± SD (n = 6).

**Figure 6 biomedicines-12-00180-f006:**
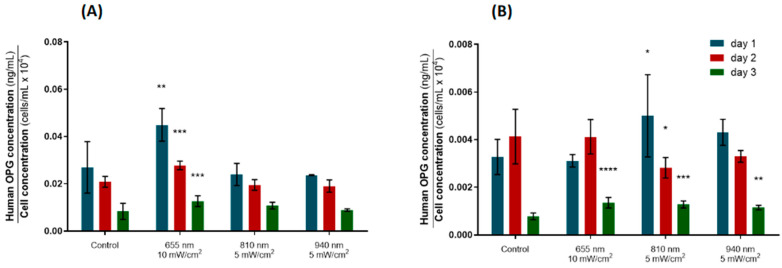
ELISA data for the determination of OPG expression for the optimal photobiomodulation parameters, normalized by cell counting. (**A**) Normalized OPG concentration for hPLFs. (**B**) Normalized OPG concentration for hFOBs. Statistically significant differences in relation to control are indicated above the highest OPG concentration value. * *p*  <  0.05; ** *p*  <  0.01; *** *p*  <  0.001; **** *p*  <  0.0001. Data are represented as mean ± SD (n = 2).

**Figure 7 biomedicines-12-00180-f007:**
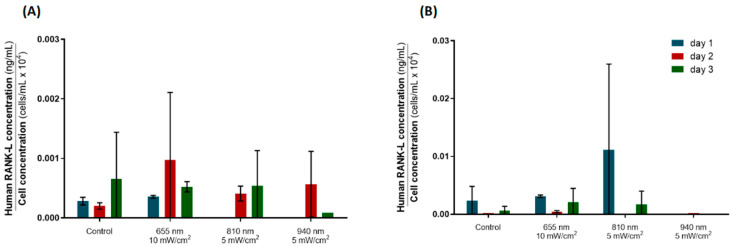
ELISA data for the determination of RANK-L expression for the optimal photobiomodulation parameters, normalized by cell counting. (**A**) Normalized RANK-L concentration for hPLFs. (**B**) Normalized RANK-L concentration for hFOBs. Statistically significant differences in relation to control are indicated above the highest RANK-L concentration value. Data are represented as mean ± SD (n = 2).

**Table 1 biomedicines-12-00180-t001:** Summary of the irradiation parameters tested in hFOBs and hPLFs.

	Wavelength (nm)	Power Density (mW/cm^2^)	Duration (min)	Number of Sessions	Follow-Ups
Initial screening	655 810 940	5 10	3	1	1, 24 and 72 h (after stimulation)
Optimal parameter testing phase				3	1, 2 and 3 days (after each stimulation)

**Table 2 biomedicines-12-00180-t002:** Summary of the main outcomes of the optimal stimulation parameters for hFOBs and hPLFs.

		Metabolic Activity ^#^	ALP Expression ^#^	OPG Expression ^#^	RANK-L Expression ^#^
**Periodontal ligament** **fibroblasts**	**↑**	655 nm at 10 mW/cm^2^ by day 2 (non-sig)	655 nm at 10 mW/cm^2^ by day 2 810 nm at 5 mW/cm^2^ by day 2 (non-sig)	655 nm at 10 mW/cm^2^ by days 1, 2 and 3	810 nm at 5 mW/cm^2^ by day 2 (non-sig)
	**↓**	-	-	-	-
**Osteoblasts**	**↑**	655 nm at 10 mW/cm^2^ by day 3 810 nm at 5 mW/cm^2^ by days 1 and 3	655 nm at 10 mW/cm^2^ by day 3 810 nm at 5 mW/cm^2^ by days 1 and 3 940 nm at 5 at 10 mW/cm^2^ by day 3	655 nm at 10 mW/cm^2^ by day 3 810 nm at 5 mW/cm^2^ by days 1 and 3 940 nm at 5 at 10 mW/cm^2^ by day 3	-
	**↓**	-	-	810 nm at 5 mW/cm^2^ by day 2	-

^#^ Normalized by cell counting. Caption: ↑: increased; ↓: decreased; non-sig: no statistically significant differences were found.

## Data Availability

The datasets used and analyzed during the current study are available from the corresponding author on reasonable request.

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
