# Peer review of "Optimization of a Photobiomodulation Protocol to Improve the Cell Viability, Proliferation and Protein Expression in Osteoblasts and Periodontal Ligament Fibroblasts for Accelerated Orthodontic Treatment"

_biomedicines, 2024, doi:10.3390/biomedicines12010180_

Round 1
Reviewer 1 Report
Comments and Suggestions for Authors
The topic of the manuscript is to assess the impact of different PBM protocols on cellular metabolic activity of PDL fibroblasts and osteoblasts, as well as on the expression of key bone remodeling mediators, including ALP, OPG and RANK-L.
The title and the abstract of the article are informative. The Introduction presents the issue of photobiomodulation and its therapeutic potential. The section "Material and Methods" precisely describes the chosen study design. The sections "Results" and “Discussion” require some revisions, e.g. the figure captions, the study limitations and more recent references should be added. The Conclusions should be more "take-home" messages.
Some following points must be clarified/corrected for the further processing of this article.
1. The abstract should be a single paragraph and should follow the style of structured abstracts but without headings.
2. The graphical abstract should probably refer directly to the results of the own research rather than present the general background with a bit of methodology.
3. In the Introduction, there are long text passages without proper references.
4. None of the Figures have any captions. They should be self-explanatory. In the case of graphs, information on the statistical tests used shall be provided.
5. The Results section contains many text and graphs without any result table that would be worth considering.
6. The Discussion must be supplemented by study limitations.
7. It is suggested to add more recent articles from 2022-2023 to the references in the Introduction and the Discussion (e.g. 10.3390/app122211532, 10.3390/metabo13040520, 10.1016/j.heliyon.2023.e13220, 10.1111/php.13787, 10.1631/jzus.B2200706).
8. The Conclusions should be more "take-home" messages.
9. References should be described as follows:
1. Author 1, A.B.; Author 2, C.D. Title of the article. Abbreviated Journal Name Year, Volume, page range.
Author Response
The topic of the manuscript is to assess the impact of different PBM protocols on cellular metabolic activity of PDL fibroblasts and osteoblasts, as well as on the expression of key bone remodeling mediators, including ALP, OPG and RANK-L.
The title and the abstract of the article are informative. The Introduction presents the issue of photobiomodulation and its therapeutic potential. The section "Material and Methods" precisely describes the chosen study design. The sections "Results" and “Discussion” require some revisions, e.g. the figure captions, the study limitations and more recent references should be added. The Conclusions should be more "take-home" messages.
Authors’ response: Dear reviewer, first of all, we would like to acknowledge the reviewer’s time and the attention given to our manuscript. We also appreciate all your helpful recommendations. The revised manuscript includes the requested modifications highlighted in yellow.
Some following points must be clarified/corrected for the further processing of this article.
- The abstract should be a single paragraph and should follow the style of structured abstracts but without headings.
Authors’ response: We would like to thank the reviewer for his/her comment. The demanded adjustments were conducted in the Abstract (page 1).
- The graphical abstract should probably refer directly to the results of the own research rather than present the general background with a bit of methodology.
Authors’ response: We acknowledge the reviewer’s comment concerning this point. We would like to highlight that the graphical abstract entirely refers to the results of the current research, and no data from previous literature is depicted. Also, the graphical abstract aims to provide a brief illustration of the methodology and the main results regarding the most suitable parameters to promote specific bone remodeling phenomena. In this sense, we believe that the presented graphical abstract provides an illustrative summary of the current research paper.
- In the Introduction, there are long text passages without proper references.
Authors’ response: We would like to thank the reviewer for this constructive comment, and we recognize that some references are missing. These were added to the revised version of the manuscript.
- None of the Figures have any captions. They should be self-explanatory. In the case of graphs, information on the statistical tests used shall be provided.
Authors’ response: We acknowledge the reviewers’ attention to this point. Indeed, the absence of all figure captions was a mistake, which was already corrected in the new version of the manuscript.
- The Results section contains many text and graphs without any result table that would be worth considering.
Authors’ response: We acknowledge the reviewer’s comment concerning this point. As suggested, we added a table depicting a summary of the main outcomes of the optimization protocol (Table 2, page 11).
- The Discussion must be supplemented by study limitations.
Authors’ response: We appreciate the reviewer’s feedback on this topic. Indeed, the former Discussion section lacked a proper consideration of the study limitations, which is now provided in the revised version of the manuscript (page 14, lines 498-505).
- It is suggested to add more recent articles from 2022-2023 to the references in the Introduction and the Discussion (e.g. 10.3390/app122211532, 10.3390/metabo13040520, 10.1016/j.heliyon.2023.e13220, 10.1111/php.13787, 10.1631/jzus.B2200706).
Authors’ response: We thank the reviewer for the highly interesting references he/she has provided. These and other recent articles (including 10.1007/s12015-021-10142-w, 10.1007/s10103-023-03870-7, and 10.1302/2046-3758.81.BJR-2018-0060.R1) were included in the reference list of the manuscript and cited throughout the text.
- The Conclusions should be more "take-home" messages.
Authors’ response: We appreciate your helpful recommendations. We recognize that the Conclusions section was too much prosaic. In the revised version of the manuscript, conclusions are presented in a more straight-forward way, as bullet points (page 14, lines 515-522).
- References should be described as follows:
1. Author 1, A.B.; Author 2, C.D. Title of the article. Abbreviated Journal NameYear, Volume, page range.
Authors’ response: Thank you for the attention given to this point. The reference list is produced by a reference manager software, and therefore the suggested corrections should be ultimately made in the final proof. We assure you that all the references will be revised and checked-out before publication to meet the journal standards.
We hope the alterations can meet the reviewer’s expectations.

Reviewer 2 Report
Comments and Suggestions for Authors
I would like to thank the authors for their efforts to conduct such an intriguing subject. The present manuscript entitled “Optimization of a photobiomodulation protocol to improve cell viability, proliferation, and protein expression in osteoblasts and periodontal ligament fibroblasts for accelerated orthodontic treatment” reports different wavelengths and two power densities to find the suitable photomodulation. While your research has potential, addressing, these issues are crucial for reconsideration.
Minor comments:
1. Improve readability by using shorter sentences, as some of the current sentences are lengthy and not easy to follow. For example, lines 71-77.
2. Please define abbreviations at least once the first time they are used, for example, PNPP.
3. Please check the consistency in font sizes throughout the text, for example, line 356.
4. The conclusion of the present study fails to emphasize the significance of the findings and lacks the necessary structure expected in a conventional conclusion.
5. All the sentences in the report should be written in passive format. There are many sentences starting by “we”. Please revise.
6. Many figures are missing a related caption.
Major comments:
Abstract
7. I believe that the most important part of a manuscript is the abstract part. It should be well-written, concise, and informative, and it should provide the reader with the most important. To improve the abstract, please provide more emphasis on the objectives and results of the study.
8. Please revise the following sentence: “In this sense, we propose an optimization protocol to find the most suitable …”. Avoid using the phrase “most suitable.”
Background
9. The literature survey is very limited, please expand the literature survey.
Method
10. Revise Table 1 to accurately reflect the results discussed in the text; it currently only presents the initial data for performance testing.
Result
11. Please, provide clear labels for each component (a, b, ….) in Figure 3 to specify what they represent.
Discussion
12. Please clarify the inconsistency between the mention of MTS in the text and the reference to MTT in the discussion section for accuracy and coherence.
13. In the Discussion section, figure numbers should correspond with their respective explanations. Please revise the relevant items. For example, the explanation of Figure 2.
14. Please add figures related to culture and cellular assay.
Comments on the Quality of English LanguageMinor editing of English language required
Author Response
I would like to thank the authors for their efforts to conduct such an intriguing subject. The present manuscript entitled “Optimization of a photobiomodulation protocol to improve cell viability, proliferation, and protein expression in osteoblasts and periodontal ligament fibroblasts for accelerated orthodontic treatment” reports different wavelengths and two power densities to find the suitable photomodulation. While your research has potential, addressing, these issues are crucial for reconsideration.
Minor comments:
- Improve readability by using shorter sentences, as some of the current sentences are lengthy and not easy to follow. For example, lines 71-77.
Authors’ response: Dear reviewer, first of all, we would like to thank you for the constructive comments and suggestions, and we acknowledge the reviewer’s time and attention given to our manuscript. The revised manuscript includes the requested modifications highlighted in yellow.
Regarding the existence of lengthy and not-easy-to-read sentences, we apologizes for this matter. This and other similar sentences were revised in the new version of the manuscript.
- Please define abbreviations at least once the first time they are used, for example, PNPP.
Authors’ response: We appreciate your attention to this point. The abbreviations were revised throughout the text to ensure that any other definition is missed.
- Please check the consistency in font sizes throughout the text, for example, line 356.
Authors’ response: We acknowledge the reviewer’s attention to this point. All formatting issues were rectified in the revised version of the manuscript.
- The conclusion of the present study fails to emphasize the significance of the findings and lacks the necessary structure expected in a conventional conclusion.
Authors’ response: We appreciate your helpful recommendations. We recognize that the Conclusions section was too much prosaic. In the revised version of the manuscript, conclusions are presented in a more straight-forward way, as bullet points (page 14, lines 514-521).
- All the sentences in the report should be written in passive format. There are many sentences starting by “we”. Please revise.
Authors’ response: We would like to thank the reviewer for his/her comment. The demanded adjustments were conducted throughout the text.
- Many figures are missing a related caption.
Authors’ response: We acknowledge the reviewers’ attention to this point. Indeed, the absence of all figure captions was a mistake, which was already corrected in the new version of the manuscript.
Major comments:
Abstract
- I believe that the most important part of a manuscript is the abstract part. It should be well-written, concise, and informative, and it should provide the reader with the most important. To improve the abstract, please provide more emphasis on the objectives and results of the study.
Authors’ response: We would like to thank the reviewer for this constructive comment, and we recognize that some references are missing. These were added to the revised version of the manuscript.
- Please revise the following sentence: “In this sense, we propose an optimization protocol to find the most suitable …”. Avoid using the phrase “most suitable.”
Authors’ response: We acknowledge the reviewers’ attention to this point. This expression was replaced by “optimal” and “most appropriate” in the new version of the manuscript.
Background
- The literature survey is very limited, please expand the literature survey.
Authors’ response: We would like to thank the reviewer for this constructive comment, and we recognize that the literature survey was quite limited. Recent references in the scope of this research paper were added to the Introduction and Discussion sections of the revised version of the manuscript, including 10.3390/app122211532, 10.3390/metabo13040520, 10.1016/j.heliyon.2023.e13220, 10.1111/php.13787, 10.1631/jzus.B2200706, 10.1007/s12015-021-10142-w, 10.1007/s10103-023-03870-7, and 10.1302/2046-3758.81.BJR-2018-0060.R1, among others.
Method
- Revise Table 1 to accurately reflect the results discussed in the text; it currently only presents the initial data for performance testing.
Authors’ response: We acknowledge the reviewer’s comment concerning this point. Table 1 provides a summary of the PBM parameters initially addressed in the current study. We added Table 2 to the manuscript, which presents the main outcomes of the optimization protocol.
Result
- Please, provide clear labels for each component (a, b, ….) in Figure 3 to specify what they represent.
Authors’ response: We appreciate the reviewer’s feedback on this topic. Indeed, figure captions were missing in the original version of the manuscript. Now, all figures present a description of the depicted results, statistical tests and significance, and all the respective labels.
Discussion
- Please clarify the inconsistency between the mention of MTS in the text and the reference to MTT in the discussion section for accuracy and coherence.
Authors’ response: We thank the reviewer for the highly interesting references he/she has provided. MTS and MTT are two distinct types of assays used to measure cell viability in vitro, which are very similar in terms of concept and methodology (MTS is an update, short version of the MTT assay). Indeed, the MTS test is commonly referred to as “one-step MTT”. A brief mention of the tests’ similarity was added to the revised manuscript.
- In the Discussion section, figure numbers should correspond with their respective explanations. Please revise the relevant items. For example, the explanation of Figure 2.
Authors’ response: We appreciate your attention to this error. All figure numbers of the Discussion were corrected in the new manuscript.
- Please add figures related to culture and cellular assay.
Authors’ response: We acknowledge the reviewer’s helpful recommendations. In this particular case, we would like to highlight that all the cellular and molecular tests are well-established and broadly used bioassays. In this sense, we believe that the inclusion of a figure depicting the bioassays used in the current study would not add considerable value to the manuscript, and it can even swamp the paper with too many figures, graphs and schemes. In addition, we would like to emphasize that we provide a graphical abstract, which provides a brief illustration of the methodology, as well as of the main results of the optimization study.
Concerning the minor editing of English language suggested by the reviewer, we would like to point out that lengthy and not-easy-to-follow sentences were revised, as well as other grammar and readability concerns.
We hope the alterations can meet the reviewer’s expectations.

Reviewer 3 Report
Comments and Suggestions for Authors
1. What are the Photobiomodulation Effects on Periodontal Ligament Stem Cells?
2. What are the Effects of LASER photobiomodulation on the cell viability of periodontal ligament fibroblasts?
3. Under which mechanism does photobiomodulation reduce oxidative stress in fibroblast cells by inhibiting the FOXO1 signaling pathway?
4. What is the optimal time-response window for photobiomodulation therapy combined with a static magnetic field?
Author Response
- What are the Photobiomodulation Effects on Periodontal Ligament Stem Cells?
Authors’ response: Dear reviewer, first of all, we would like to thank you for the constructive comments and suggestions, and we acknowledge the reviewer’s time and attention given to our manuscript. The revised manuscript includes the requested modifications highlighted in yellow.
Regarding your comment on the effect pf PBM on periodontal ligament stem cells, briefly, photobiomodulation can enhance the differentiation capacities of PDL stem cells, improve their proliferation and suppress an excessive inflammatory response cause, for instance, by a mechanical stress. A short insight on the ability of PBM to modulate PDL stem cells osteogenic or osteoclastic differentiation was added to the Introduction (page 3, lines 92-96).
- What are the Effects of LASER photobiomodulation on the cell viability of periodontal ligament fibroblasts?
Authors’ response: We appreciate your attention to this point. Indeed, the initial manuscript provided information on the effect of PBM on cell viability of osteoblasts, but not of periodontal ligament fibroblasts (and stem cells – described above). Indeed, few in vitro studies in PDL fibroblastic cultures were conducted so far, which is now highlighted in the Introduction of the revised manuscript (page 3, lines 86-89 & page 12, lines 418-419).
- Under which mechanism does photobiomodulation reduce oxidative stress in fibroblast cells; by inhibiting the FOXO1 signaling pathway?
Authors’ response: We acknowledge the reviewer’s attention to this point. However, describing the signaling pathways by which PBM decreases oxidative stress was not the main focus of the current optimization protocol, and therefore we have only considered the including of a brief description on how light interacts with the mitochondria and increases ATP production. WE suggest the paper AIMS Biophys. 2017 ; 4(3): 337–361. doi:10.3934/biophy.2017.3.337 for further insights on this topic. Nevertheless, we rectified the description of the mechanisms of action of PBM, specifically in the production of ROS, in the Introduction (page 2, line 63-69).
- What is the optimal time-response window for photobiomodulation therapy combined with a static magnetic field?
Authors’ response: We acknowledge the suggestions provided by the reviewer, which were all carefully considered for the improvement of our manuscript. In this specific issue, we understand the reviewer’s point, but we would like to point out that the combination of PBM with other stimulation modalities was not in the scope of the current research paper. Nevertheless, based on the results reported in the current manuscript, we can anticipate that the optimal time-response window to maximize the acceleration of bone remodeling phenomena by PBM depends on the type of light protocols applied: i) daily PBM of 655 nm delivered at a power density of 10 mW/cm2, as well as irradiation of 810 and 940 nm delivered at 5 mW/cm2 and applied for three consecutive days to promote bone formation; and ii) PBM of 810 nm at 5 mW/cm2 applied for two consecutive days and suspended by the third day, to facilitate osteoclastogenesis and, hence, bone resorption.
The combination of PBM with alternative therapeutic modalities may be an interesting feature to unveil in forthcoming research, which is now recommended in the revised version of the manuscript (page 15, lines 548-552).
In addition, the reviewer pointed out that the results could be more clearly described and that the bridge from such results to the conclusions must be better explained. In the revised version of the manuscript, we have tried to clarify and highlight the main results of the current optimization protocol. Among other changes, we have added a table summarizing the main outcomes of the study (Table 2, page 11), and we now present the main conclusions as bullet points, to highlight the principal take-home messages. We hope the alterations can meet the reviewer’s expectations.

Round 2
Reviewer 1 Report
Comments and Suggestions for Authors
The Authors responded to practically all the comments, significantly improving the manuscript. Thank you, and congratulations.
The only minor thing is that the bibliography style still does not correspond to the publisher's guidelines.
Author Response
Dear reviewer,
We would like to acknowledge all your helpful recommendations, which significantly improved our manuscript.
Since this should be the last version of the manuscript, we have now updated the reference list according to the publisher’s guidelines.
Reviewer 2 Report
Comments and Suggestions for Authors
Dear Authors, Thank you for the revised version.
Author Response

(The authors gave the same response as above.)
